# Study on the Degradation of Aflatoxin B1 by *Myroides odoratimimus* 3J2MO

**DOI:** 10.3390/biology14060724

**Published:** 2025-06-19

**Authors:** Xue Wang, Yao-Yao Gao, Dun Wang, Qi Zhang, Hao-Ran Wang, Ting-Ting Zhang, Meng-Jie Zhu, Jing Dong, Dong Ling, Peng Feng, Xue-Hui Tang, Pei-Wu Li

**Affiliations:** 1Xiangyang Academy of Agricultural Sciences, Xiangyang 441057, China; maysxuer@126.com (X.W.); wanghaoran_12@163.com (H.-R.W.); 18327852733@163.com (T.-T.Z.); fengpengxynky@163.com (P.F.); tangxh19782023@163.com (X.-H.T.); 2Hubei Hongshan Laboratory, Wuhan 430061, China; 3Oil Crops Research Institute, Chinese Academy of Agriculture Science, Wuhan 430062, China; gaoy4743@gmail.com; 4Key Laboratory for Biotoxin Detection, Ministry of Agriculture and Rural Affairs, Wuhan 430062, China; 5College of Food Science and Engineering, Wuhan Polytechnic University, Wuhan 430062, China

**Keywords:** aflatoxin, helicase, peroxidase, biodegradation

## Abstract

Researchers isolated *Myroides odoratimimus* (3J2MO) from the soil, demonstrating the effective degradation of multiple aflatoxins. The enzymes, using biochemical purification, synergistically degrade AFB1 in contaminated peanuts and reduced AFB1 by 93.8% within 24 h. This enzymatic approach offers an efficient, chemical-free solution for mitigating aflatoxin contamination in agriculture and food systems.

## 1. Introduction

Aflatoxin (AFT) is a secondary metabolite produced by *Aspergillus* spp. The common and well-studied chemical structures are aflatoxin B1 (AFB1), aflatoxin B2 (AFB2), aflatoxin G1 (AFG1), aflatoxin G2 (AFG2), aflatoxin M1 (AFM1), etc. [1,2]. Aflatoxin groups B and G are widely found in peanuts, maize, rice, wheat, nuts, and feeds [3,4], whereas aflatoxin M1 is usually found in animal tissues, animal bodily fluids (such as milk and urine) and dairy products [5,6,7,8]. AFB1 is the most structurally stable and toxic compared to other aflatoxins. AFB1 is more prevalent in animal feeds and AFB1-infested grains and can still be detected after 20 years of storage [9,10].

*Aspergillus flavus* may infest crops at different stages of growth, harvest, transport, storage, and processing. During these processes, differences in environmental factors such as light, temperature, water activity, hosts, etc., can affect aflatoxin production, and environmental conditions such as humidity and temperature are very likely to lead to aflatoxin production [11,12,13]. Aflatoxin contamination in China has obvious geographical characteristics [14]. Among them, the main peanut-producing areas in the south have the most serious aflatoxin contamination. China’s exports of peanuts have repeatedly been impacted by aflatoxin contamination [15].

Therefore, it is of great significance to study detoxification methods for aflatoxins. The decomposition temperature of aflatoxin is around 268 °C, so it has strong heat resistance [16,17]. Traditional methods (e.g., chemical adsorption and thermal treatment) are limited by toxin persistence and nutrient destruction. The advantages of aflatoxin removal methods, especially biological methods, which are efficient and safe have always been a topic of interest [18]. Based on their mechanism, these can be divided into two types, biosorption and biodegradation methods. Bioenzymatic degradation for detoxification is an effective strategy to reduce aflatoxin concentrations in food and feed [19]. Compared with traditional physicochemical methods, detoxification using microorganisms or bioenzymes is characterized by mild conditions and a good detoxification effect, does not damage the nutritional quality, and does not require complex equipment [20].

Due to the diversity of *Aspergillus flavus* and the toxins it produces in the environment and the uncertainty of the impact of different microbial detoxification methods applied to the environment, fewer microbial detoxification methods are actually used in production practice. Our team performed a study on *Myroides odoratimimus* 3J2MO [21]. The strain was found to have the ability to efficiently degrade a wide range of toxins and is one of the strains found to efficiently degrade a wide range of biotoxins. Strain screening is the testing of the AFB1 degradation of fermented food isolates by HPLC. Enzyme purification was performed by successive steps (ammonium sulfate precipitation and ion-exchange chromatography) to separate the active fractions. The enzyme mechanism was then analyzed and the detoxification safety evaluation was assessed by in vivo experiments in mice.

Therefore, this paper analyzed and identified the active components of AFB1-degrading enzymes in the fermentation broth, and continued the research on the degradation mechanism of aflatoxin B1 by *Myroides odoratimimus* 3J2MO on the basis of the team’s previous research, aiming to provide new materials for the biological control of AFB1 and laying the theoretical foundation for the creation of a further method for the biological detoxification of AFB1.

## 2. Materials and Methods

### 2.1. Preparation of Degradation Matrix and Optimization of Conditions

The test strain *Myroides odoratimimus* 3J2MO was provided by the Key Laboratory of Biotoxin Detection of the Ministry of Agriculture, Institute of Oilseed Crops, Chinese Academy of Agricultural Sciences (CAAS, (Wuhan, China)). The original strain was stored in a −80 °C refrigerator, and the test strain was stored in a −20 °C refrigerator. *Myroides odoratimimus* 3J2MO was activated in LB solid medium by a plate streaking method, and cultured for 24 h at 37 °C in a constant-temperature incubator. A single colony of *Myroides odoratimimus* 3J2MO was picked from the plate, transferred to 50 mL liquid Luria-Bertani (LB) medium (Sigma-Aldrich, St. Louis, MO, USA), and incubated at 37 °C on a 200 rpm shaker (Shanghai, China). The growth liquid was taken at different incubation times, and the optical density (OD 600) value of the liquid was measured at 600 nm with enzyme labelling equipment (Molecular Devices, San Jose, CA, USA). The OD value was regularly measured at 100 h. The growth curve of *Myroides odoratimimus* 3J2MO was plotted with the incubation time as the horizontal coordinate and the OD value of the bacterial solution as the vertical coordinate. From this curve, the growth trend of the strain was analyzed and the peak growth period was derived.

A single colony of activated *Myroides odoratimimus* 3J2MO was picked and transferred to 50 mL of LB liquid medium and incubated for 24 h at 37 °C and 200 rpm with shaking. On the following day, at a 5% addition, the culture was transferred to 1000 mL of LB liquid medium and incubated at 37 °C and 200 rpm, with shaking for 24 h. The supernatant, bacterial suspension, and intracellular fluid of *Myroides odoratimimus* 3J2MO were prepared according to the following methods, respectively. The 3J2MO supernatant was obtained by placing the 3J2MO fermentation broth in a centrifuge and centrifuging it at 10,000 rpm for 20 min at 4 °C. The bacterial suspension was obtained from 3J2MO fermentation broth, centrifuged at 10,000 rpm and 4 °C for 20 min. The centrifuged bacteria were washed with sterile water, repeated 3 times; centrifuged at 10,000 rpm and 4 °C for 20 min; and resuspended in 50 mL (20 mmol/L Tris-HCl), resulting in the bacterial suspension. The above suspension of *Myroides odoratimimus* 3J2MO was broken by low-temperature ultrasonication (3 s of work, 8 s of interval, 350 W, and 30 min) and centrifuged at 10,000 rpm and 4 °C for 20 min; the supernatant was collected and filtered with a 0.22 µm filter membrane. This is how the intracellular fluid was obtained. In order to analyze active components in the fermentation broth of *Myroides odoratimimus* 3J2MO, the following procedure was performed. First, 990 µL of supernatant, bacterial suspension, and bacterial intracellular fluid were transferred into a 1.5 mL centrifuge tube. Next, 10 µL of AFB1 (Sigma, USA) standard solution (final concentration: 100 µg/L) was added to each tube, followed by incubation at 37 °C (200 rpm) for 72 h in a constant-temperature shaker. Five technical replicates were prepared under identical conditions to ensure reproducibility. The control groups were sterile LB liquid medium (supernatant) and 20 mmol/L Tris-HCl (Aladdin, Shanghai, China) solution (bacterial suspension and intracellular fluid). The reaction solutions of the three groups were examined by high-performance liquid chromatography (HPLC (Agilent1100, Santa Clara, CA, USA)) for the content of AFB1 [22].

To further clarify the type of active ingredients in the supernatant of *Myroides odoratimimus* 3J2MO, 990 µL of treatment solutions 1, 2, 3, and 4 were added to 1.5 mL centrifuge tubes. To each, 10 µL of AFB1 standard solution was added, reaching a final concentration of 100 µg/L. The samples were then placed in a constant-temperature shaker, incubated at 37 °C and 200 rpm for 72 h. Five parallels were set for each treatment group. The control group consisted of the blank supernatant. The supernatant of *Myroides odoratimimus* 3J2MO was subjected to the following four treatments. Processing methodology I involves the following steps: Take 1 mL of the supernatant, add 0.01 g of 1% (sodium dodecyl sulfate) SDS solution (Aladdin, Shanghai, China), and put it into a 60 °C water bath for 2 h. Processing methodology 2 involves the following steps: Take 1 mL of the supernatant, add 1 mg of proteinase K, and place it in a 60 °C water bath for 2 h. Processing methodology 3 entails the following steps: Take 1 mL of supernatant, add 0.01 g of a 1% SDS solution and 1 mg of proteinase K (Absin (Shanghai, China)) simultaneously, and place it in a 60 °C water bath for 2 h. Processing methodology 4 follows these steps: Take 20 mL of supernatant and autoclave at 100 °C for 20 min. After each of the above treatments, the samples were filtered through a 0.22 µm organic filtration membrane, and the content of AFB1 was detected by HPLC.

### 2.2. Purification and Identification of AFB1-Degrading Enzyme

Following the centrifugation of the fermentation broth of *Myroides odoratimimus* 3J2MO, the supernatant was collected. A total of 1000 mL of supernatant was transferred, adjusted to pH 7.4 with 1 M Tris-HCl, and divided into 100 mL aliquots. Ammonium sulfate was incrementally added to the supernatant under constant stirring to avoid protein denaturation caused by rapid salting-out. To the fermentation supernatant (4 °C), solid ammonium sulfate was added incrementally to achieve final saturation levels of 40%, 50%, 60%, 65%, 70%, and 80% (*w*/*v*). Each addition was performed gradually under constant stirring at 4 °C to avoid localized supersaturation. The mixture was incubated overnight at 4 °C, followed by centrifugation at 10,000× *g* for 20 min at 4 °C. The resulting precipitates were collected by decanting the supernatant. From each of the above solutions with different saturation levels, 990 µL was taken, 10 µL of AFB1 standard solution (final concentration 100 µg/L) was added, placed in 1.5 mL centrifuge tubes, and incubated at 200 rpm for 72 h at 37 °C in a thermostatic shaker, and the experimental group was set up with five replications. The content of AFB1 was detected by HPLC and the degradation rate of AFB1 was calculated to determine the most suitable conditions for ammonium sulfate precipitation. The crude enzyme solution post-dialysis was transferred to a 50 mL centrifuge tube and centrifuged at 10,000× *g* for 10 min to remove debris. The protein concentration was quantified using the BCA Protein Assay Kit (Thermo Fisher Scientific, Waltham, MA, USA) following the manufacturer’s protocol. For protein analysis, samples were resolved by SDS-PAGE (12% polyacrylamide gel) under reducing conditions and visualized via Coomassie blue staining [23].

The crude enzyme solution was separated and purified using AKTA Pure (Thermo Fisher Scientific, USA), and the ion-exchange chromatography column (Q-Sepharose Fast Flow (Cytiva, Sweden), column volume 25 mL) was linearly eluted with a mobile phase containing 1 mol/L NaCl, and the eluted peaks were collected using centrifugal tubes [24], with 5 mL collected from each tube. The solution from each tube, which was collected under the same peak, was desalted using 35 kDa ultrafiltration tubes, freeze-dried, and re-solubilized. From each of the above mentioned tubes of the complex solution, 990 µL of each solution was added to 10 µL of AFB1 standard solution (final concentration 100 µg/L), placed in 1.5 mL centrifuge tubes, and incubated in a constant-temperature shaker at 37 °C at 200 rpm for 72 h. The content of AFB1 was detected by HPLC, and the rate of the degradation of AFB1 was calculated to determine the degradation activity of the peaks in each of the collected tubes. Among them, the most active component was the active enzyme solution. The active enzyme purified by anion-exchange chromatography was purified by gel filtration chromatography [25,26]. The eluate of each component was collected, freeze-dried, and re-dissolved separately, 990 µL of solution was taken and assayed, and the component with the strongest degradation activity was the target enzyme solution [27].

Precipitation experiments using different saturation degrees of ammonium sulfate show (Table 1) that 80% of saturated ammonium sulfate precipitation of protein content is the highest, with a content of 0.5925 mg. In addition, when the ammonium sulfate saturation degree continues to increase, the precipitation of protein content also gradually increases, between 40 and 80%; the ammonium sulfate precipitation of protein content shows an increasing trend [28].

The purified “target enzyme solution” was separated by electrophoresis using denaturing SDS-PAGE, stained with Coomassie blue, and used to cut the gel to recover the target protein bands. The bands in the SDS-PAGE gel were cut off using a sterile scalpel. After decolorization, CAA, TCEP solution was added and incubated at 60 °C for 30 min. Then, acetonitrile was added, followed by vortexing for 5 min, centrifuging and discarding the supernatant, vacuum drying, trypsin, and 37 °C incubation with shaking, overnight. Then, peptide extraction solution (ACN/formic acid) was added, followed by ultrasonic extraction for 10 min, and centrifugation to obtain the supernatant; this was sent to UW Genetics for mass spectrometry [29]. Mass spectrometry results were retrieved using MaxQuant (V1.6.6.0) software, and the data algorithm was adopted from Andromeda. The reference databases used for the search were the custom and NCBI databases.

### 2.3. Bioinformatics Experiment Design

The search parameters were as follows: fixed modification carbamidomethyl (C); variable modification oxidation (M); acetyl (protein N-term); and trypsin/P for digestion. The primary mass spectrometry matching tolerance was set at 20 ppm for the initial search, and the secondary mass spectrometry matching tolerance was also set at 20 ppm, with the primary search set at 4.5 ppm. The search results were filtered by 1% FDR at the protein and peptide level, and entries of contaminated proteins, anti-column proteins, and proteins with only one modified peptide were deleted. The remaining identification information was used for subsequent analyses [30,31]. The *Myroides odoratimimus* 3J2MO strain was sent to Shanghai Sangon Biotech Co., Ltd. (Shanghai, China) for sequencing. The 16S rRNA coding sequence of the *Myroides odoratimimus* 3J2MO strain was registered on the National Center for Biotechnology Information (NCBI) website (accession number: PQ656451), and was used as a query sequence for BLAST (BLAST+ 2.14.0) analysis in *Myroides odoratimimus* genomes available on NCBI. Seventy-two 16S rRNA coding sequences with high homology with the 3J2MO strain were selected and downloaded. Meanwhile, eight 16S rRNA coding sequences from representative strains which have been reported to have AFB1-degradation activities were also downloaded and used for cluster analysis. After alignment, all 16S rRNA coding sequences from a total of 81 strains were used to construct a phylogenetic tree using the neighbor-joining (NJ) method in MEGA-X (MEGA 6) software (bootstrap replications of 1000 and a partial detection threshold of 50).

### 2.4. Data Analysis

The degradation rate of AFB1 was calculated using the following formula. Degradation rate % = (1 − A/A0) × 100% (A is the peak area of AFB1 in the sample; A0 is the peak area of AFB1 in the control). Each experiment was performed in triplicate and provided consistent results with SPSS 20 software. Graphing was performed using origin 7.5. Assay results were retrieved using MaxQuant (V1.6.6.0) software for mass spectrometry information, and the algorithm for the data was Andromeda. The search results were screened based on 1% FDR at the protein and peptide levels, and contaminating proteins, reverse library proteins, and protein entries with only one modified peptide were deleted, and the remaining identification information was used for subsequent analysis.

GO/COG/KEGG annotation was used to understand the function of the degradative enzymes.

The physicochemical properties, hydrophilicity analysis, and prediction of the protein tertiary structure of amino acid sequences were obtained using online software (Table 2).

## 3. Results

### 3.1. Determination of Degradation Conditions

The growth curve of *Myroides odoratimimus* 3J2MO was plotted (Figure 1a). It can be seen that the strain enters the logarithmic growth phase after 4 h, and the organism in the logarithmic growth phase grows vigorously and divides actively. In 23–31 h, the growth of the strain is in the stable phase, which can produce a large number of secondary metabolites. After 31 h, the growth of the strain enters into the decay phase. Therefore, the cultivation time with OD value around 1.2 was selected as the best inoculation period in this experiment. To improve the experimental reproducibility, the degradation conditions were clarified. The buffer composition was 50 mM Tris-HCl (pH 7.4) with 5 mM MgCl_2_ and 100 μg/mL BSA. The temperature was 37 °C. The reaction was completed within 15 min to avoid substrate depletion. The final substrate concentration of AFB1 was 10 μM. We carried out AFB1 degradation rate experiments on the supernatant, bacterial suspension, and intracellular fluid of the fermentation broth of *Myroides odoratimimus* 3J2MO after co-cultivation with AFB1, respectively. The results showed (Figure 1b) that the active substances in the supernatant had the best degradation effect on AFB1, with a degradation rate of up to 95%. The active substances that degrade AFB1 in *Myroides odoratimimus* 3J2MO are enzymes or other extracellular metabolites produced by the bacterium’s metabolism and secreted into the supernatant. As shown in Figure 1c, the degradation rate of the supernatant after treatment with SDS and proteinase K was at 30%. In addition, the degradation rate of AFB1 after co-treatment with SDS and proteinase K was even lower, with a degradation rate of only 15%; the degradation capacity of the supernatant for AFB1 was as high as 90% in comparison with that of untreated supernatant. Therefore, it was hypothesized that the active substances for the degradation of AFB1 by *Myroides odoratimimus* 3J2MO were extracellular enzymes secreted by the bacterium into the supernatant.

### 3.2. Determination of Degradative Function

Through the determination of the distribution of the degradation component of the strain, the degradation conditions and degradation rate of the component will be screened and used for comparative analysis. AFB1 degradation rate experiments revealed the change rule of AFB1-degrading enzyme activity in the precipitation fractions of ammonium sulfate with different saturation degrees (Figure 2a). The saturation degree of ammonium sulfate was between 60 and 80%, the precipitated protein was positively correlated with the degradation rate of AFB1, and 80% saturated ammonium sulfate precipitated protein had the best ability to degrade AFB1, with a degradation rate of 80%. Therefore, 80% saturated ammonium sulfate was chosen as the best condition for precipitating proteins.

After the ammonium sulfate precipitation experiment, the crude enzyme solution was further separated and purified using a Q-Sepharose Fast Flow chromatography column. Plotting outflow curves (Figure 2b), a total of four peaks A, B, C, and D were collected, and the protein solution of each peak collected by ion exchange, respectively, was co-cultured with AFB1 for 72 h, and its degradation rate of AFB1 was detected (Figure 2c). Among them, the peak B protein solution had the strongest ability to degrade AFB1 with a degradation rate of 75%. Therefore, peak B protein solution was selected as the main degradation active ingredient and isolated and purified by gel filtration chromatography. The results are shown in Figure 2d. After separation by gel filtration chromatography, two elution peaks were separated by UV detection at 280 nm, labelled as peak 1 and peak 2. The protein solutions collected from each peak were tested for AFB1 degradation activity, as shown in Figure 2e, and it was found that the degradation of peak 1 protein solution was the highest, with a degradation rate of 64% for AFB1. Therefore, it can be surmised that the AFB1-degrading enzyme is mainly present in peak 1. The molecular weight of AFB1-degrading enzyme was detected by SDS-PAGE gel electrophoresis. As shown in Figure 2f, the crude enzyme solution had the most protein bands, and it was impossible to distinguish the target bands. It was hypothesized that the band could be the target protein that plays a major degradation role, and the molecular weight of the protein was predicted to be around 45 kDa based on marker. target 3, labelled in blue in Figure 2f.

We refined the enzyme purification protocol (Figure 2g) to enhance recovery, purity, and detection sensitivity (Tukey’s test was used to compare the differences between groups). By adjusting the ammonium sulfate salting-out gradient to 40–80% and employing Q-Sepharose column chromatography, we achieved a >95% pure enzyme preparation. Fluorescence detection at 280 nm further streamlined activity monitoring, enabling the precise quantification of the target enzyme during purification.

Following this optimization, we rigorously evaluated the biosafety of the purified enzyme using in vivo toxicity assays. In compliance with NY/T 1109-2017 [32], acute toxicity tests including intraperitoneal injection (0.1 mL/10 g-BW), oral administration, and percutaneous exposure were conducted at a maximum dose of 500 mg/kg-BW. Both the fermentation broth of *Myroides odoratimimus* 3J2MO and the AFB1 degradation products exhibited no mortality or adverse effects in mice after 14 days, confirming their safety for potential applications.

### 3.3. Bioinformatics Analysis

By completing the determination of the degradation ability of the strains, the similarity of the degradation ability of other strains will be further comparatively analyzed; therefore, we analyzed the homology of *Myroides odoratimimus* by gene matching. An evolutionary tree was constructed based on the coding sequence of 16S rRNA (Figure 3). We found that eight bacteria reported in the literature with AFB1-degrading activity were located at the root of the tree. The 3J2MO (PQ656451) strain (marked in red) clustered on a large branch with 72 *Myroides odoratimimus* strains whose genome data were publicly available on NCBI. This indicates that they are significantly different from bacteria of several other groups (*Pseudomonas*, *Lactobacillus*, and *Bacillus*) with aflatoxin-degrading activity in terms of biological classification. In addition, within the genus *Myroides odoratimimus*, there are multiple branches. The J2MO (PQ656451) strain is on one of the small branches together with KC172018 *Myroides odoratimimus*, OM956401 *Myroides odoratimimus*, JF775419 *Myroides odoratimimus*, and PP239584 *Myroides odoratimimus*. This suggests the presence of variations among different strains of *Myroides odoratimimus*.

### 3.4. Functional Analysis of Degrading Enzymes

Based on the previous analysis of the strain’s ability to degrade toxins, it was inferred that the degradation was due to the presence of proteases capable of degradation in the bacterial fermentation broth. The nature of the degrading enzyme is further analyzed and characterized below. The screening of characteristic peptides for quantitative protein detection included analyzing the corresponding match score and interference number of theoretical enzymatic peptides of proteins to be tested by BLAST comparison, and selecting the theoretical enzymatic peptides with a match score of ≥20 and an interference number of ≤5 as the candidate peptides. This time, one protein gel strip was identified by mass spectrometry, and the protein-level mass spectrometry information is shown in Figure 4a, and the number of unique peptides (Unique) greater than one or equal to one was selected as the significant result [33]. The information on the number of selected unique peptides was compared with BLAST from custom database and NCBI, and two valid proteins were identified, with molecular weights of 50.839 kDa and 55.983 kDa, respectively (Table 3).

The amino acid sequence of 3J2MOGL000150 is as follows:

MNTFEQFNLPKALEKALNELNIISPTPIQAKSFPVILSGRDMMGIAQTGTGKTFAYLLPILKQWKFSHAESPRVVILVPTRELVVQVVDEVEKLTAYMSVRTLGVYGGTNINTQRKAVYEGVDILVGTPGRMMDLALDGVLRFDNLQKLVIDEFDEILNLGFRTQLTSILTMMKGKRQNILFSATMTEEVDEVLDEYFDFPEEVSLAPSGTPLENIDQQIYNVPNFNTKLNLLMHLLSNKEEFNRVLIFINSKRLADVVMEKLDAAFPEEFTVIHSNKSQNFRMRSMAEPKEEEVKLAAEILMEKELRAMTIXXXXXXXXXXXXXXXXXXXXXXXXXXXXXXXXXXXXXXXXXXXXXXXXXXXX (confidential).

The amino acid sequence of 3J2MOGL003364 is as follows:

MKEQKLTTASGRPYVDHEDSLTAGARGPVLLEDYILHEKLAHFNRERIPERIVHAKGSGAYGTFTVTNDITKYTKAKLFSEVGKQTDVFVRFSTVGGEKGSADSERDPRGFAVKFYTEDGNWDLVGNNTPVFFIKDAKKFPDFIHTQKRHPGTNLKSPTMVWDFWSLNPESLHQVLILMSDRGTPFGFRHMNGYGSHTFSMINSENERVFVKFHFKTAQGIKNLTGPEADQMRATDMDYAQRDLYENISNGNFPKWNLKIQVMTEQQAKEAVNPFDVTKVWPXXXXXXXXXXXXXXXXXXXXXXXXXXXXXXXXXXXXXXXXXXXXXXXXXXXXXXXXXXXXXXXXXXXXXXXXXXXXX (confidential).

The validated proteins identified were described by the GO database (Table 4) [34], and after searching, it was found that the protein of 3J2MOGL003364 has peroxidase activity and the protein of 3J2MOGL000150 has helicase activity. The functional classification of the GO database is shown in Figure 4b. Five proteins are involved in binding, namely adenyl nucleotide binding, carbohydrate binding, ion binding, nucleotide, and nucleic acid binding. Five are biologically active, mainly ATP-dependent activity, catalytic activity acting on RNA, helicase activity, hydrolase activity, and peroxidase activity.

A protein search of 3J2MOGL003364 through the COG database revealed peroxidase activity. The protein of 3J2MOGL000150 was found to have helicase and hydrolase activities by database comparison (Table 5). In addition, a comparison of the COG and GO databases revealed the similarity of the results, with both finding that the protein of 3J2MOGL000150 has helicase activity. The protein of 3J2MOGL003364 was found to have the same consistency of results by comparison of the two databases, where peroxidase is the hallmark enzyme of the peroxisome, which accounts for about 40% of the total amount of peroxisomal enzymes [35]. The results of the COG database annotation of proteins (Figure 4c) show that there are three categories, which are involved in cellular processes and signaling functions for cell wall/membrane/envelope biogenesis and resistance mechanisms. Involved in information storage and processing functions are replication reorganization and repair. Metabolism-related functions are energy generation and transformation [36].

The KEGG database (Table 6) described two validated proteins: the protein of 3J2MOGL003364 was found to have peroxidase activity [EC:1.11.1.6] and the protein of 3J2MOGL000150 was searched and found to have ATP-dependent RNA helicase enzyme activity [EC:3.6.4.13]. The KEGG database is functionally annotated (Figure 4d), with different colors indicating different pathways, and there are six categories. Four of them are involved in metabolism, namely secondary metabolite synthesis, carbon metabolism, tryptophan metabolism, and glyoxylate and dicarboxylic acid metabolism. All of them are related to environmental information processing: the Fox0 pathway, the plant-MAPK signaling pathway, and the yeast-MAPK signaling pathway, respectively. Three of them have functions related to human diseases, namely neurodegenerative pathways in various diseases, chemical carcinogenesis by reactive oxygen species, and amyotrophic lateral cord sclerosis. Three functions involved in organic systems of organisms are the longevity regulation pathway, longevity regulation pathway of multiple species, and longevity regulation pathway of worms. Involved in the processing of genetic information is the degradation of RNA, and the function associated with the cellular process is the peroxisome. The KEGG database is more detailed in annotating the functions of proteins, showing more metabolic pathways [37,38].

By comparison of GO, COG, and KEGG databases, among them, the protein of 3J2MOGL000150 has helicase enzyme activity; the protein of 3J2MOGL003364 has peroxidase activity. Therefore, it can be preliminarily hypothesized that AFB1-degrading enzymes have helicase and peroxidase activities.

In order to study the efficient degradation mechanism of the protease, we analyzed the hydrophilic and hydrophobic properties of the two enzymes and made a preliminary analysis of the three-dimensional spatial structure of two enzymes (Figure 5a,b). The results showed that both formed a hydrophobic interface through the N-terminal structural domain. A synergistic mechanism is hypothesized between the two enzymes, with peroxidase oxidizing AFB1 to generate free radical intermediates, and helicase providing energy to facilitate the decomposition of the intermediates through ATP hydrolysis. However, according to recent studies, a resolution of 0.44 Å can be achieved using chromatographic techniques, which validates our structural docking simulation (ΔG = −12.3 kcal/mol) as a complementary approach. Given the limitations of sample stability under EM conditions (less than 5% retention of AFB1 degradation products after 2 h at 80 kV accelerating voltage), we plan to opt for computational modeling to predict the mechanism of peroxidase–helicase interaction [39,40]. The N-terminal structural domain stabilizes the complex through hydrophobic interaction by 3D spatial structure analysis (Figure 5c,d). The dehydrogenase provides energy through ATP hydrolysis to drive electron transfer from the peroxidase active center. The model can provide a theoretical basis for designing efficient AFB1-degrading enzyme engineering.

## 4. Discussion

The enzyme method reduced aflatoxin-related medical costs in the test area. The production of compliant groundnut in Senegal would allow exports to increase from 25,000 to 210,000 tons, with an increase of >USD 300 million in annual revenue [41]. It meets the limits of the European Union EC No 1881/2006 (AFB1 < 2 μg/kg), but FDA approval is required for the U.S. market. Since our isolated enzyme was less effective in degrading other toxins (mainly vomitoxin, etc.), further studies are needed to design enzyme variants for wider applications and to monitor the residual enzyme activity in the soil for 3 years to assess the ecological footprint. The mechanistic pathways of enzyme synthesis will be designed in later stages of the study, including the cleavage of the lactone ring by the enzyme, laccase cross-linking of metabolites, and the dynamics of biocontrol agent colonization. The enzymatic digestion of aflatoxins has progressed through a deeper understanding of mechanisms and scalable field applications. Future work should focus on enzyme immobilization and regulatory compliance to bridge the gap between lab-scale effects and practical applications.

In our study, the aflatoxin degradation products were confirmed by mass spectrometry, and the 3J2MO enzyme destroyed the lactone ring of AFB1 by free radical oxidation. This is consistent with recent findings on the activity of bacteria in degrading aflatoxins [42,43,44]. Enzyme-mediated degradation products have been found to increase water solubility by the degradation of active ingredients, a mechanism also employed in the degradation of other aflatoxins [45]. Enzyme treatments reduced the bioavailability of soil heavy metals by 58%, whereas chemical methods exacerbated contamination [46]. This illustrates the safety of the bioenzymatic degradation method. In terms of broad-spectrum toxin degradation, it was found that in a maize trial, the bioenzymes degraded 85.6% of the biotoxins, which exceeded most microbial agents (e.g., *Aspergillus niger*: 62.3%) [7,41,47]. It was shown that all of these degrading enzymes were broad-spectrum, readily produced and obtained in bacteria, and functioned in a relatively similar manner to the enzymes produced by bacteria, such as *Pseudomonas* and *Bacillus*, which suggests their origin in soil microbiota [8,48,49]. The degradation potential of 3J2MO enzymes needs to be further explored.

Studies have found that catalase reduces the damage of oxidative stress to cells by decomposing hydrogen peroxide into water and oxygen [10,50]. Research has shown that catalase can scavenge H_2_O_2_ generated during toxin metabolism, thus protecting cells from oxidative damage [51]. This protective role is exemplified in AFM1 degradation, where catalase synergizes with other enzymes to reduce toxicity. Notably, catalase also facilitates oxidative cyclization via superoxide anion generation, broadening its biotechnological utility in detoxification [52]. RNA helicases, conversely, regulate toxin metabolism indirectly by modulating RNA splicing and repair, potentially alleviating cellular damage through RNA degradation [53]. While these enzymes operate through distinct pathways, their synergistic potential such as catalase targeting oxidizing agents and RNA helicases repairing toxin-induced RNA damage remains underexplored [54]. Recent advances further highlight engineered peroxidases (e.g., Rh_DypB N246A mutant) that efficiently degrade AFB1 by hydroxylating intermediates like AFQ1, achieving 96% bioconversion at low enzyme/hydrogen peroxide concentrations [55]. Such findings underscore the importance of enzymatic synergy in detoxification, setting the stage for our investigation into *Myroides odoratimimus* 3J2MO.

Our study reveals that peroxidase exhibits superior specificity in toxin degradation compared to helicase. Structural analysis demonstrates that peroxidase’s hydrophobic interface functionally complements the ATP-binding domain of deconjugate enzymes, driving stable complex assembly. Crucially, the spatiotemporal coordination of electron transfer (via Fe^3+^/Fe^2+^ redox cycling) and ATP hydrolysis emerges as the linchpin for synergistic degradation. These mechanistic insights align with prior observations of peroxidase efficiency (e.g., MnP-mediated AFB1 oxidation to stable dihydrodiol products [43]) but uniquely demonstrate how *Myroides odoratimimus* 3J2MO enzymes achieve irreversible, high-yield detoxification through modular enzyme cooperation. This work provides a theoretical framework for designing robust bioremediation systems, though further studies are needed to optimize enzyme stability and scalability in real-world applications.

## 5. Conclusions

Our team first reported the AFB1 degradation activity of *Myroides odoratimimus* 3J2MO in 2019 [16]. To further understand the molecular mechanism of AFB1 degradation by this strain, this study conducted an in-depth analysis of *Myroides odoratimimus* 3J2MO through biochemical, molecular biological, and evolutionary biological methods. This study successfully optimized the conditions for the degradation of aflatoxin by microbial degradation enzymes, and biological methods provide a new idea for the efficient degradation of aflatoxin. By analyzing the composition of degradation components, this study demonstrated the high efficiency of biodegradation enzymes in degrading aflatoxins. In addition, the degradation product composition was analyzed by mass spectrometry methods, which provided valuable information for an in-depth understanding to study the safety of this degradative enzyme preparation. Functional and structural studies of the degrading enzyme revealed the key functions of peroxidase and the deconjugate enzyme, and the synergistic effect of the two enzymes has important potential. In conclusion, this study lays the foundation for future toxin degradation technology by complex enzyme preparations and for exploring methods for the green degradation of toxins in grain, oil, and feed matrices.

## 6. Outlook

In the previous experiments [56], crude enzyme solution was added to peanut meal contaminated by AFB1, and AFB1 in peanut meal was effectively degraded, but there are still some problems, which need to be resolved with in-depth research. Mycotoxin biodegradation technology is one of the more advanced technologies with the least impact on feed nutrition and the best detoxification effect, and it is especially necessary to study, develop, and promote the use of mycotoxin biodegradation technology in feed. In the later stage of development, biological agents targeting peroxidase or RNA helicase will be developed. Advances in technological tools such as single-cell sequencing and CRISPR screening should also be utilized to precisely regulate the activity of these enzymes. Through genetic engineering, microorganisms will be modified to secrete peroxidase–helicase complexes, which can degrade hidden toxins and provide theoretical support and technical breakthroughs for the industrial biomanufacturing of antitoxins.

## Figures and Tables

**Figure 1 biology-14-00724-f001:**
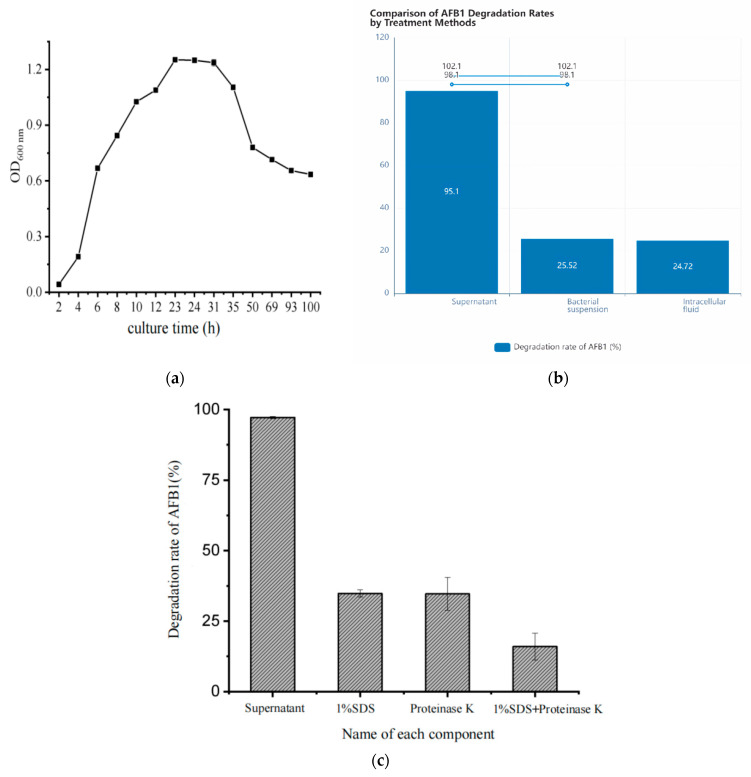
Determination of degradation conditions. Growth curves of different groups (e.g., experimental vs. control) were compared by repeated measures analysis of variance (RM-ANOVA, *p* < 0.01). (**a**) Growth curve of *Myroides odoratimimus* 3J2MO. (**b**) Comparison of the degradation rate of AFB1 by each component of the fermentation broth. (**c**) The effect on the degradation rate of AFB1 by treating the crude enzyme solution with different denaturation methods.

**Figure 2 biology-14-00724-f002:**
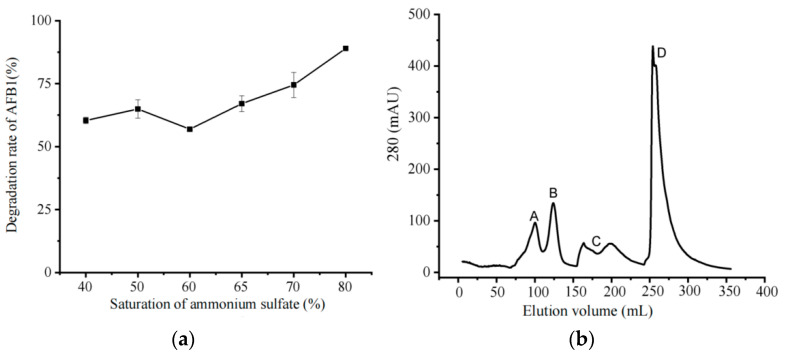
Enzyme purification. Using repeated measures ANOVA (RM-ANOVA, *p* < 0.01), group and time effects were significant, and further differences in degradation rates across groups at specific groups and time points were compared by Tukey et al. (**a**) Effect of crude enzyme solution treated with different saturations of ammonium sulfate on the degradation rate of AFB1. (**b**) Ion-exchange chromatography purification results. Note: The crude enzyme solution was separated and purified by a Q-Sepharose Fast Flow chromatography column, and the four components were named peak A, peak B, peak C, and peak D, respectively. (**c**) AFB1 degradation rate of peak A, B, C, and D in ion-exchange chromatography. (**d**) Fluorescence values of two enzyme components, components 1 and 2, separated by gel filtration chromatography. (**e**) AFB1 degradation rate of peak 1 and 2. (**f**) Electropherogram of purified degradative enzyme. Note: M denotes the protein molecular weight marker (kDa); 1, crude enzyme solution; 2, ion-exchange chromatography peak B; and 3, gel filtration chromatography peak 1. (**g**) Enzyme purification process.

**Figure 3 biology-14-00724-f003:**
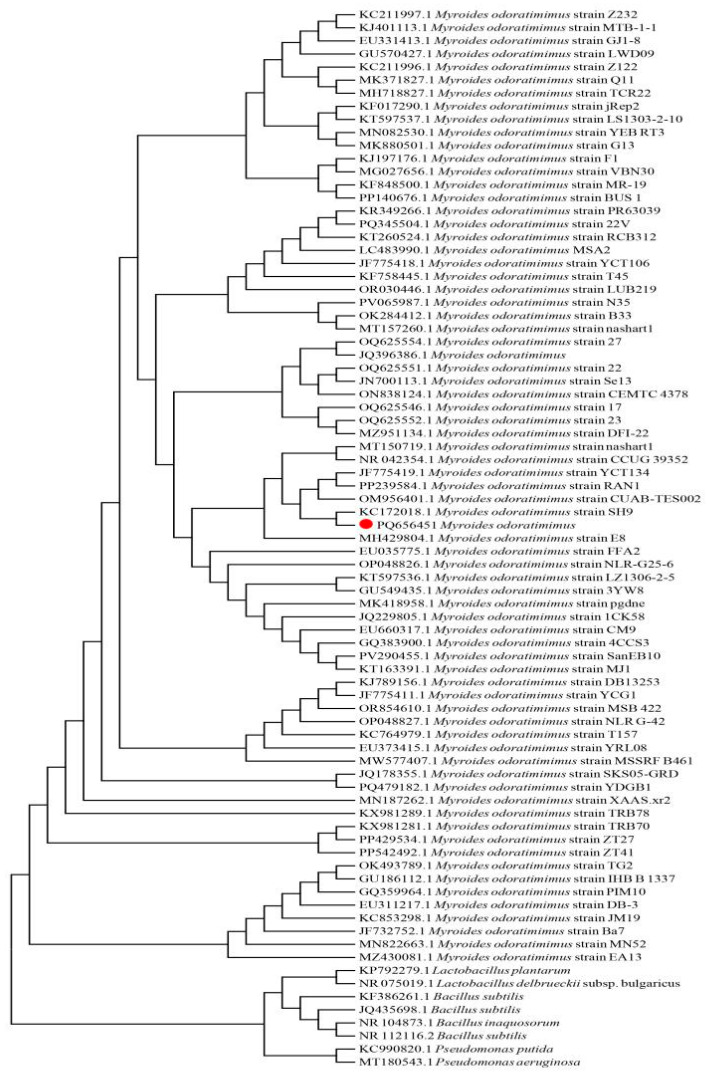
Phylogenetic analysis of strain 3J2MO based on 16srRNA sequences. (● represents the target strain).

**Figure 4 biology-14-00724-f004:**
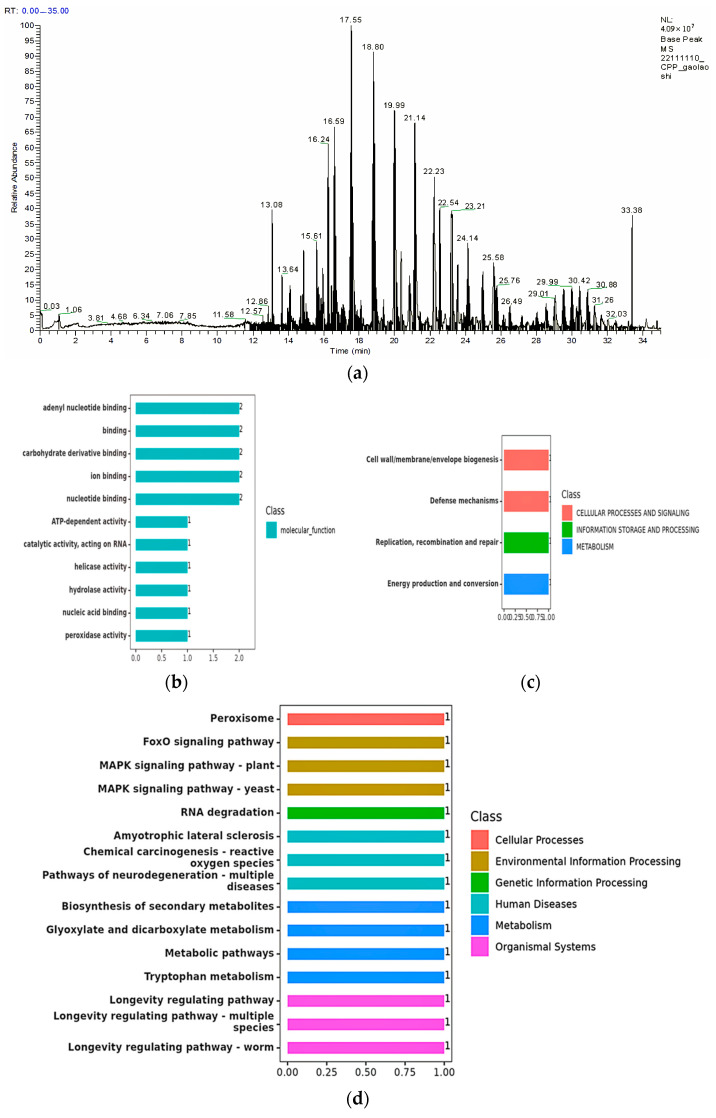
Enzyme functional analysis. (**a**) Protein primary mass spectrometry. (**b**–**d**) GO, COG, and KEGG functional classification diagrams.

**Figure 5 biology-14-00724-f005:**
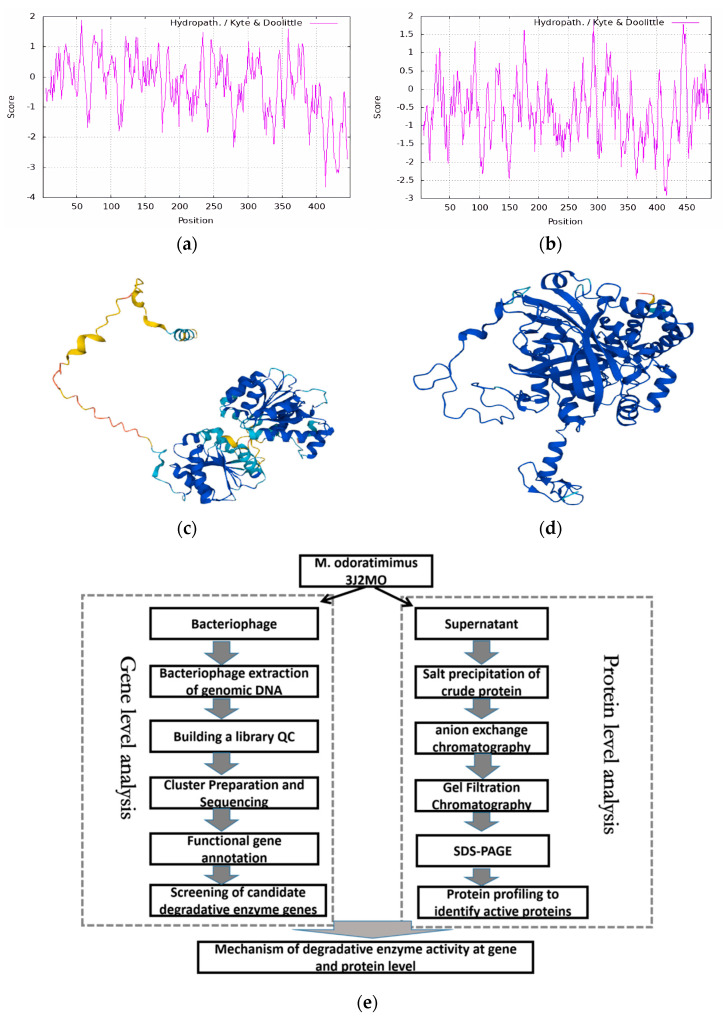
Analysis of the structure of enzymes. (**a**) 3J2MOGL000150 analysis of protein hydrophilicity and hydrophobicity. (**b**) 3J2MOGL003364 analysis of protein hydrophilicity and hydrophobicity. (**c**) 3J2MOGL000150 tertiary structure. (**d**) 3J2MOGL003364 tertiary structure. (**e**) Overall process of 3J2MO degradation enzyme mechanism method study.

**Table 1 biology-14-00724-t001:** Status of protein content precipitated by ammonium sulfate with different saturation degrees.

Ammonium sulfate saturation (%)	40	50	60	65	70	80
Protein precipitation (mg)	0.339	0.345	0.4205	0.4665	0.5235	0.5925

**Table 2 biology-14-00724-t002:** Software for analyzing biological information.

Analyze Content	Software	Website
Physical and chemical properties analysis	ProtParam 8.5.6	http://web.expasy.org/protparam/ accessed on 22 May 2023.
Hydrophilicity and hydrophobicity analysis	Protscale	http://web.expasy.org/protscale/ accessed on 11 March 2023.
Tertiary structure prediction	AlphaFold 3	https://alphafold.ebi.ac.uk/ accessed on 11 March 2023.

**Table 3 biology-14-00724-t003:** Identification results of protein glue strips.

Proteins	Mw (kDa)	Peptides	Coverage	Unique	iBAQ (%)
3J2MOGL000150	50.839	1	2.4%	1	83.3995%
3J2MOGL003364	55.983	1	2.2%	1	9.1731%

**Table 4 biology-14-00724-t004:** GO database comparison results.

Proteins	Mw (kDa)	Length	GO Description
3J2MOGL000150	50.839	449	helicase activity
3J2MOGL003364	55.983	492	peroxidase activity

**Table 5 biology-14-00724-t005:** Results of COG database comparison.

Proteins	Mw (kDa)	Length	COG Description
3J2MOGL000150	50.839	449	helicase activity and hydrolase activity
3J2MOGL003364	55.983	492	peroxidase activity

**Table 6 biology-14-00724-t006:** KEGG database comparison results.

Proteins	Mw (kDa)	Length	KEGG Description
3J2MOGL000150	50.839	449	rhlE; ATP-dependent RNA helicase RhlE [EC:3.6.4.13]
3J2MOGL003364	55.983	492	katE, CAT, catB, srpA; catalase [EC:1.11.1.6]

## Data Availability

Data available on request from the authors.

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
