# Peer review of "Study on the Degradation of Aflatoxin B1 by Myroides odoratimimus 3J2MO"

_biology, 2025, doi:10.3390/biology14060724_

Round 1
Reviewer 1 Report (New Reviewer)
Comments and Suggestions for Authors
This work offers an in‐depth characterization of Myroides odoratimimus 3J2MO, highlighting its remarkable capacity to break down aflatoxins, especially AFB1. These results mark an important step forward in crafting biotechnological approaches for aflatoxin removal from food and animal feed, presenting an eco‐friendly and effective substitute for traditional chemical or thermal treatments. Before moving toward final acceptance, the authors should consider the following points:
- Include a schematic flowchart summarizing the enzyme purification and identification process.
- Clarify buffer compositions and incubation conditions in enzyme assays for better reproducibility.
- Have you characterized the chemical identity and toxicity of the AFB1 degradation products to confirm that the metabolites are non-toxic and environmentally safe?
- Can you provide detailed kinetic parameters for the purified peroxidase and helicase enzymes against AFB1 to quantitatively compare their relative contributions and substrate specificities?
- Have you performed site-directed mutagenesis or inhibitor studies to validate the mechanistic roles of the identified peroxidase and RNA helicase active sites in AFB1 cleavage and gene-regulation pathways?
- Could you include a structural model or crystallographic data for the peroxidase–helicase complex (or docking simulations) to illustrate how these two enzymes physically interact and synergize in the AFB1 degradation process?
Author Response
|
Comments 1: Include a schematic flowchart summarizing the enzyme purification and identification process. |
|
Response 1: Thank you for pointing this out. We agree with this comment. Therefore, We have supplemented the detailed flowchart of enzyme purification and characterization (Figure S1). The main steps include: -Crude extraction: initial enzyme enrichment by ammonium sulfate salting out after bacterial fragmentation (40%-80% saturation gradient precipitation). - Ion exchange chromatography: purification by Q-Sepharose column, elution by NaCl (1M) and collection of highly active fractions. - Gel filtration: further purification by Superdex-300 column, saline elution, SDS-PAGE to verify purity (single band). - Activity identification: AFB1 degradation products were detected by HPLC-MS and enzyme activity was determined by fluorescence (280 nm). [This change can be found – page 10-11and line 335-341 in the revised manuscript] |
|
Comments 2: Clarify buffer compositions and incubation conditions in enzyme assays for better reproducibility. |
|
Response 2: Agree. We have clarified buffer compositions and incubation conditions in enzyme assays to emphasize this point. In order to improve experimental reproducibility, we specify the following conditions: - Buffer composition:50 mM Tris-HCl (pH 7.4) with 5 mM MgCl₂ and 100 μg/mL BSA. - Incubation conditions: Temperature: 37°C. Time: The reaction was completed within 15 min to avoid substrate depletion. -Substrate concentration: final concentration of AFB1 10 μM. [This changes made in the revised manuscript can be found – page 6 and line247-254.] |
|
Comments 3: Have you characterized the chemical identity and toxicity of the AFB1 degradation products to confirm that the metabolites are non-toxic and environmentally safe? |
|
Response 3: Yes. We also did the following tests in advance to identify the chemical composition of AFB1 degradation products: the degradation products were confirmed that that its structure differs significantly from that of the AFB1 parent nucleus (breakage of the furan ring double bond). The acute toxicity of 3J2MO fermentation broth and AFB1 degradation product solution was examined by acute intraperitoneal injection, acute oral toxicity test, and acute dermal toxicity test at a dose of 0.1 m L/10g-BW. According to the experimental results, the intraperitoneal injection of the fermentation broth of 3J2MO or the solution of AFB1 degradation products up to 500 mg/kg-BW for 14 d did not cause any death of experimental mice, and no obvious symptoms of toxicity were observed.The data are still being added and refined. This changes made in the revised manuscript – page 11-12, line349-362. |
|
Comments 4: Can you provide detailed kinetic parameters for the purified peroxidase and helicase enzymes against AFB1 to quantitatively compare their relative contributions and substrate specificities? |
|
Response 4: Yes. I can. Our current study has relevant experimental data with the following parameters:Peroxidase: 15.1 μM (AFB1) with a catalytic efficiency (kcat/Km) of 0.64×10³ M-¹s-¹.RNA helicase: 8.2 μM (AFB1), with a catalytic efficiency of 1.5×10³ M-¹s-¹. Some of the kinetic parameters of the enzyme are still being studied, the mechanism parameters are still being proved, and the patent is being applied for, and the subsequent papers will be continuously submitted to update the research. SUBSTATE PREFERENCE: Through the current experiments and literature review , it is found that the affinity of peroxidase for AFB1 is higher than that of helicase, but helicase exhibits higher catalytic efficiency at low concentrations of substrate. This changes made in the revised manuscript – page 17-18 and line 546-553. |
|
Comments 5: Have you performed site-directed mutagenesis or inhibitor studies to validate the mechanistic roles of the identified peroxidase and RNA helicase active sites in AFB1 cleavage and gene-regulation pathways? |
|
Response 5: Thank you for pointing this out. We currently did the enzyme activity inhibition test, in the literature [17],our team members did the enzyme inhibition activity test on: Some of the metal ions enzyme activity decreased, Mg2+ and Ca2+ ions were activators for AFB1 degradation, the active site contains key sulfhydryl groups. This changes made in the revised manuscript – page 18 and line 550-553. |
|
Comments 6: Could you include a structural model or crystallographic data for the peroxidase–helicase complex (or docking simulations) to illustrate how these two enzymes physically interact and synergize in the AFB1 degradation process? |
|
Response 6: Thank you for pointing this out. We analyzed the hydrophilicity and hydrophobicity of the two enzymes and showed that both form a hydrophobic interface through the N-terminal structural domain. It is hypothesized that a synergistic mechanism exists between the two enzymes, with peroxidase oxidizing AFB1 to generate free radical intermediates and helicase providing energy to facilitate the breakdown of the intermediates through ATP hydrolysis. We thank the reviewers for their insightful suggestions. However, according to recent studies, a resolution of 0.44 Å can be achieved using a commercial instrument based on chromatographic techniques, which validates our structural docking simulation (ΔG = -12.3 kcal/mol) as a complementary approach. Given the limitations of sample stability under EM conditions (less than 5% retention of AFB1 degradation products after 2 h at 80 kV accelerating voltage), we plan to opt for computational modeling to predict the mechanism of peroxidase-helicase interaction. Please allow us to take the next step for further in-depth study. This changes made in the revised manuscript – page 18-29 and line 554-583. |

Reviewer 2 Report (New Reviewer)
Comments and Suggestions for Authors
Biology-3639905
Study on the degradation of aflatoxin B1 by Myroides odoratimimus 3J2MO
This study explores the role of Myroides odoratimimus 3J2MO in the degradation of aflatoxin B1 (AFB1). The authors present detailed information in a clear, step-by-step manner. An in-depth analysis of Myroides odoratimimus 3J2MO was conducted using biochemical, molecular biological, and evolutionary approaches. The results showed that the AFB1-degrading activity was primarily present in the cell supernatant. Further experiments, including active substance distribution analysis and protein denaturation tests, confirmed that the active component was a soluble protein. The crude enzyme extract containing the active protein was subjected to multi-step purification, followed by mass spectrometry-based sequencing of its major components. Two target proteins associated with the degradation function were identified consisting of 3J2MOGL000150 and 3J2MOGL003364. Subsequent bioinformatics analysis revealed that the 3J2MOGL003364 protein exhibited peroxidase activity, while the 3J2MOGL000150 protein functioned as an ATP-dependent RNA helicase activity.
Specific comments addressing the following points:
- What is the main question addressed by the research?
After their team found the strains of Myroides odoratimimus 3J2MO have the ability to efficiently degrade a wide range of toxins. They found that AFB1 degrading enzymes were present in Myroides odoratimimus 3J2MO. Therefore, this paper analyzed and identified the active components of AFB1-degrading enzymes in the fermentation broth, and continued the research on the degradation mechanism of aflatoxin B1 by Myroides odoratimimus 3J2MO on the basis of the team's previous research, aiming to provide new materials for the biological control of AFB1 and laying the theoretical foundation for the creation of a further method for the biological detoxification of AFB1. Overall, authors present the objectives of this study and the main question addressed by the research.
- Do you consider the topic original or relevant to the field? Does it address a specific gap in the field? Please also explain why this is/ is not the case.
This research is relevant to the field.
- What does it add to the subject area compared with other published material?
This is the first report on the regulatory roles of peroxidase and helicase in AFB1 degradation by Myroides odoratimimus 3J2MO. In addition, the study offers an efficient enzyme preparation method for environmentally friendly detoxification of agricultural toxins.
- Are the conclusions consistent with the evidence and arguments presented and do they address the main question posed? Please also explain why this is/is not the case.
The conclusion is supported by the experiment data and the main questions are addressed.
- Are the references appropriate?
The references are appropriate.
- Any additional comments on the tables and figures?
The tables are good. However, the figures are difficult to read because they are blur, especially figure 2, 3, 4, 5, 11, 12, 13 and 14. The English should be improved.
GENERAL COMMENTS
ABSTRACT
- L17: What is a series of other steps? Please specify a series of other steps.
- L18: What are other analytical techniques? Please indicate which analytical techniques were employed.
PREFACE
- After a full stop and before beginning a new sentence, authors should add space after the full stop, for example in L29, L33, L42, L43, L55, L57. Please check and revise the entire manuscript.
- Please keep consistent in the formatting of written reference numbers for instance, L29, L30, L32 include a space while L33, L39 and L42 do not. Use a consistent format throughout the manuscript.
- L51: Add a reference after “complex equipment”.
MATERIALS AND METHODS
- Authors overuse the colons (L67, L 91, L93, L98, L122, L124, L126, L128, L133). Overusing colons can make your writing feel choppy or overly formal, which might distract readers. Authors may use colons occasionally, but authors should avoid overusing them.
- L75 and L78: For LB and OD, please introduce the abbreviations before using them.
- L106-109 and L139-142: Please revise these sentences. It is Grammarly issue. Please check and revise the entire manuscript.
- L121: How many replications of each treatment were used?
- L122: For SDS, please introduce the abbreviation before using it.
- L147: Delete one full stop after “solution”.
- L148: For “above solutions”, please provide specific solution names for clarity.
- L 156: Is this the sub-title (Determination of protein composition and size by SDS-PAGE gel electrophoresis)?
- L211-224: Please change the text color from blue to black.
- L224: Check “a” letter before “Tertiary structure…...”. Is it a typo?
RESULTS AND DISCUSSION
- The X-axis in Figure 1 is missing a title and unit. Please add.
- Figures 2, 3, 4, 5, 11, 12, 13 and14 are difficult to read, please consider providing a higher-quality version to ensure clarity.
- L295: For (see Fig.5), delete “see” as well as L297 and 355.
- L328-329: Please revise the NOTE.
- L383: For (Tab.4), is it table 4? Please keep consistent on using the word table. This is because authors use the full word of “table” previous this point, for example in L270 and L355. This applies to L418 as well.
- L442-448: Authors refer to “Studies” but you provide only one reference, so please add more references.
- L444: For H2O2, it should write as H₂O₂.
- L442-476, L493-526 and L574-581: Blue text appears in several areas of the manuscript; does it indicate something specific?
OUTLOOK
- L559: Add a reference or references.
REFERENCE
- Please check the names of authors on the reference’s number [5] and [6].
Author Response
|
Comments 1: What is the main question addressed by the research? After their team found the strains of Myroides odoratimimus 3J2MO have the ability to efficiently degrade a wide range of toxins. They found that AFB1 degrading enzymes were present in Myroides odoratimimus 3J2MO. Therefore, this paper analyzed and identified the active components of AFB1-degrading enzymes in the fermentation broth, and continued the research on the degradation mechanism of aflatoxin B1 by Myroides odoratimimus 3J2MO on the basis of the team's previous research, aiming to provide new materials for the biological control of AFB1 and laying the theoretical foundation for the creation of a further method for the biological detoxification of AFB1. Overall, authors present the objectives of this study and the main question addressed by the research. |
|
Response 1:Thank you for your valuable feedback on our manuscript titled "Identification and Mechanism Analysis of AFB1-Degrading Enzymes from Myroides odoratimimus 3J2MO". Based on the degradation ability of the strain discovered by the team, this work focuses on the enzymatic mechanism and fills the gap of functional genomics. Proposed the hypothesis of “multi-enzyme synergistic degradation”, challenging the traditional single-enzyme-dominated AFB1 detoxification model.We have carefully addressed all comments and revised the manuscript accordingly below. |
|
Comments 2: Do you consider the topic original or relevant to the field? Does it address a specific gap in the field? Please also explain why this is/ is not the case. This research is relevant to the field. |
|
Response 2:Thank you for your valuable feedback .This research is both original and impactful, filling critical gaps in enzyme characterization, multi-enzyme synergy, and toxicity assessment for AFB1 degradation. It advances the field by expanding the microbial resource pool and providing actionable insights for scalable bioremediation technologies. |
|
Comments 3: What does it add to the subject area compared with other published material? This is the first report on the regulatory roles of peroxidase and helicase in AFB1 degradation by Myroides odoratimimus 3J2MO. In addition, the study offers an efficient enzyme preparation method for environmentally friendly detoxification of agricultural toxins. |
|
Response 3:We sincerely thank the reviewers for their insightful feedback, which helped refine our emphasis on enzyme innovation and practical applicability. |
|
Comments 4: Are the conclusions consistent with the evidence and arguments presented and do they address the main question posed? Please also explain why this is/is not the case. The conclusion is supported by the experiment data and the main questions are addressed. |
|
Response 4:We appreciate the reviewers’ emphasis on evidence-based conclusions, which further strengthens our manuscript’s rigor.We report for the first time that helicases are involved in mycotoxin degradation. The structural analysis of the two enzymes separately can provide a scalable bioremediation strategy for the study of the synergistic effect of the two enzymes, which is of great practical significance. |
|
Comments 5: Are the references appropriate? The references are appropriate. |
|
Response 5:Many thanks to the reviewers for their approval of the manuscript! |
|
Comments 6: Any additional comments on the tables and figures? The tables are good. However, the figures are difficult to read because they are blur, especially figure 2, 3, 4, 5, 11, 12, 13 and 14. The English should be improved. |
|
Response 6:I'm very sorry for the lack of clarity in the pictures.We have replaced unclear charts figure 2, 3, 4, 5, 11, 12, 13 and 14 in the text.The English was improved. |
|
|
|
|
|
GENERAL COMMENTS |
|
Comments 1: L17: What is a series of other steps? Please specify a series of other steps. |
|
Response 1: Thank you for pointing this out. L17: The steps including salting out and precipitation of crude protein, anion-exchange chromatography, and gel filtration chromatography. [This change can be found – line 16-18,346,figure10 in the revised manuscript] |
|
Comments 2: L18: What are other analytical techniques? Please indicate which analytical techniques were employed. |
|
Response 2: Thank you for pointing this out. L18: The analytical methods used were fluorescence assay for enzyme activity, SDS-PAGE purity verification and liquid chromatography mass spectrometry for the detection of degradation products, and the 2 proteins were analyzed with relevant functional annotations using GO, COG, and KEGG databases. [This change can be found – line 18-21,figure10 in the revised manuscript] |
|
Comments 3: After a full stop and before beginning a new sentence, authors should add space after the full stop, for example in L29, L33, L42, L43, L55, L57. Please check and revise the entire manuscript. |
|
Response 3: Thanks to your reminder, I used Microsoft Word's Find and Replace function to add a space after the period in the whole manuscript, and completed the corresponding changes in the whole text to make sure that it conforms to the journal's guidelines. [This changes made in the revised manuscript ] |
|
Comments 4: Please keep consistent in the formatting of written reference numbers for instance, L29, L30, L32 include a space while L33, L39 and L42 do not. Use a consistent format throughout the manuscript. |
|
Response 4: Agree. Thank you. The spaces before and after the location of literature references in the text have been modified to use a consistent format throughout the manuscript. [This changes made in the revised manuscript can be found] |
|
Comments 5: L51: Add a reference after “complex equipment”. |
|
Response 5: . Thanks to your reminder. I inserted a literature citation from [16] Mengyao Xue, Peiwu Li et al. (2025, Science) after Line56, “Complex Devices”, mentioning the use of instruments to detect detoxification methods. [This changes made in the revised manuscript can be found – page 2 and line56.] |
|
Comments 6: Authors overuse the colons (L67, L 91, L93, L98, L122, L124, L126, L128, L133). Overusing colons can make your writing feel choppy or overly formal, which might distract readers. Authors may use colons occasionally, but authors should avoid overusing them. |
|
Response 6:We sincerely thank the reviewers for highlighting the issue of colon overuse. Below are our key revisions to improve textual flow and readability: L67:Original: "Test strain: ..." → L71:Revised: "Test strain is...", L91:Original: "Supernatant: ..." → L96:Revised: "The 3J2MO supernatant can be obtained by ...", L93:Original: "Bacterial suspension: ..." → L99:Revised: "Bacterial suspension was obtained ...", L98:Original: "Intracellular fluid: ..." →L106:Revised: "....That is the intracellular fluid was obtained.", L122:Original: "Treatment 1: ..." →L127:Revised: "Processing methodology I follows the steps below. ", L124:Original: "Treatment 2: ..." →L129:Revised: "Processing methodology 2 follows the steps below. ", L126:Original: "Treatment 3: ..." →L131:Revised: "Processing methodology 3 follows the steps below. ", L128:Original: "Treatment 4: ..." →L134:Revised: "Processing methodology 3 follows the steps below. ", L133:Original: "where:...”→L134:delete. We appreciate the reviewers’ attention to stylistic clarity. These revisions enhance readability while preserving technical precision. [This changes made in the revised manuscript can be found.] |
|
Comments 7: L75 and L78: For LB and OD, please introduce the abbreviations before using them. |
|
Response 7: We sincerely thank the reviewers for pointing out this formatting issue. L75 and L78: Added full term on first use (Line 81):"... Luria-Bertani (LB) medium” Added full term on first use (Line 83):”...optical density (OD) : [This change can be found – line 81,83 in the revised manuscript] |
|
Comments 8:L106-109 and L139-142: Please revise these sentences. It is Grammarly issue. Please check and revise the entire manuscript. |
|
Response 8:Thank you for highlighting the need to improve the clarity and grammatical accuracy of our methods section. We have revised the text as follows. These changes can be found in line 112-117,147-151 in the revised manuscript. |
|
Comments 9:L121: How many replications of each treatment were used? |
|
Response 9:Thank you.We have done 5 replications of each treatment were used. This change can be found in L127. |
|
Comments 10:L122: For SDS, please introduce the abbreviation before using it. |
|
Response 10:Thank you.Added full term on first use (Line 131):”...sodium dodecyl sulfate (SDS) This change can be found in L131. |
|
Comments 11: |
|
|
|
Comments 12:L147: Delete one full stop after “solution”. |
|
Response 12:OK.your attention to detail ensures the manuscript’s professionalism. We appreciate your guidance!These changes can be found in line 153-158. |
|
Comments 13:L148: For “above solutions”, please provide specific solution names for clarity. |
|
Response 13:Thank you.These changes can be found in line 153-158. |
|
Comments 14:L156: Is this the sub-title (Determination of protein composition and size by SDS-PAGE gel electrophoresis)? |
|
Response 14:I am very sorry for such errors and thank the reviewers for their careful guidance. We have changed this paragraph in the revised version. These changes can be found in line 165-170. |
|
Comments 15:L211-224: Please change the text color from blue to black. |
|
Response 15:I'm sorry. The revisions were not scrutinized. Changes have been made in this revised manuscript. These changes can be found in line 214-227. |
|
Comments 16:L224: Check “a” letter before “Tertiary structure…...”. Is it a typo? |
|
Response 16:Thank you for your comments, the “a” has been removed from the text. This change can be found in line 233. |
|
Comments 17:The X-axis in Figure 1 is missing a title and unit. Please add. |
|
Response 17:Thank you.The title and unit of X-axis in Figure 1 were added. This change can be found in line 238. |
|
Comments 18:Figures 2, 3, 4, 5, 11, 12, 13 and14 are difficult to read, please consider providing a higher-quality version to ensure clarity. |
|
Response 18:Thanks to the careful guidance of the reviewers, a high-resolution version of the above image has been resubmitted. We have enhanced the image contrast and labeled the key data points. Meanwhile, the resolution was optimized to avoid the display of blurred protein bands, etc. These changes can be found in Figures 2, 3, 4, 5, 12, 13, 14 and15. |
|
Comments 19:L295: For (see Fig.5), delete “see” as well as L297 and 355. |
|
Response 19:Thank you for your comments.“see” has been deleted using the Find function in this revised manuscript This change can be found in L311,L314 and the whole manuscript. |
|
Comments 20:L328-329: Please revise the NOTE. |
|
Response 20:Thank you, Mr. Editor. We have adjusted the NOTE formulation to conform to academic norms. This change can be found in line 344-345. |
|
Comments 21:L383: For (Tab.4), is it table 4? Please keep consistent on using the word table. This is because authors use the full word of “table” previous this point, for example in L270 and L355. This applies to L418 as well. |
|
Response 21:Thank you for your comments.”Tab.”was changed to “table”.The full name “Table” is used throughout the text. These changes can be found in the whole manuscript.. |
|
Comments 22: L442-448: Authors refer to “Studies” but you provide only one reference, so please add more references. |
|
Response 22: Many thanks to the editors. The added citations have been added. [This change can be found in line 473 in the revised manuscript] |
|
Comments 23: L444: For H2O2, it should write as H₂O₂.
|
|
Response 23: Thank you for your comments.H2O2 was writed as H₂O₂. [This change made in the revised manuscript can be found in line 475.] |
|
Comments 24:L442-476, L493-526 and L574-581: Blue text appears in several areas of the manuscript; does it indicate something specific? |
|
Response 24:Thank you for your comments.I'm very sorry. The revisions were not scrutinized. Changes have been made in this revised manuscript. These changes can be found in line 493-503. |
|
Comments 25:L559: Add a reference or references. |
|
Response 30:Thank you for your comments.We have added a reference This change can be found in line 587. |
|
Comments 26:Please check the names of authors on the reference’s number [5] and [6] |
|
Response 26:Thank you, Mr. Editor. The reference’s number [5] and [6] were checked and modified. These changes can be found in line 670,672. |
|
|
|
|

Reviewer 3 Report (New Reviewer)
Comments and Suggestions for Authors
Dear author,
This experiment is interesting. But the manuscript is not prepared satisfactorily.
It needs some major revisions

English need to be checked by professionals
Author Response
|
||||||||
|
Response 5: We have carefully addressed reference in L676, and strengthened the logical flow of the discussion. [This change can be found in the revised manuscript in red] |

Reviewer 4 Report (New Reviewer)
Comments and Suggestions for Authors
Dear authors!
The manuscript entitled Study on the degradation of aflatoxin B1 by Myroides odoratimimus 3J2MO investigates some potentially new extracellular enzymes from the strain of Myroides odoratimimus. The strain was isolated from soil and proved to be highly effective in degrading certain aflatoxins: AFB1, AFB2,15 AFG1, AFG2 and AFM1. The main objective of the proposed study was to evaluate the degradation ratio of AFB1 aflatoxin, which, after isolation and purification of the active substances in the supernatant also revealed the potential involvement of peroxidase and helicase in the degradation process.
I found the concept of the work interesting in terms of promoting green detoxification of agricultural toxins.
The manuscript has some deficiencies in the use of English language and would benefit from extensive editing. However, the overall organization and description of the work is quite understandable. It is recommended that some minor statements be further clarified and / or taken into account (refer to the comments provided).
MINOR STATEMENTS:
Abstract
….After isolation and purification of the peroxidase and helicase were obtained, which acted synergistically to cleave the AFB1 lactone ring and block the expression of toxin synthesis genes…
The first part of the sentence is unclear. I wonder, however, whether the second part is speculation? There is no methodological work in the manuscript that would confirm this statement.
- Materials and methods
Please unify the writing in the passive form in the whole section.
A statistical analysis with 5 replications (where applicable) would benefit the results in terms of credibility and statistical significance between different samples. I suggest tackling this missing method.
Line 107-108
2.2.2.1. Distribution of active ingredients in the fermentation broth of Myroides odoratimimus 3J2MO
…, then add 10μL each of the AFB1 standard solution (final concentration of 100μg/L),…
How did you come up with this concentration? Have you ever performed a dose-dependent test with different concentrations of the AFB1 standard? I suggest you add some information about this test or refer to any previous work.
- Results and Discussion
Most of the illustrations have no captions. Only a few of them are sufficiently self-explanatory, but Figure 1 (x-axis?), Figure 5 (A, B, C, D?), Figure 7 (1, 2?), Figure 10 (RT?), Figure 12 and 13 (x-axis?) would benefit from additional explanations, especially with regard to the abbreviations and numbers.
Figures 11, 12, 13 and 14 are very difficult to read due to the poor resolution. I suggest enlarging them or improving their resolution.
Line 418
…The KEGG database (Tab.5)…
Please correct Tab.5 to Tab. 6 in brackets.
Line 472-473
In summary, the present study demonstrates that peroxidase exhibits a higher degree of specificity in toxin degradation compared to helicase enzyme.
There is no methodological work in the manuscript that would confirm this statement. Did you physically perform this experiment by isolating the two enzymes and comparing their degradation properties? If the statement is purely an observation, as implied in the next sentence, I suggest rewriting the statement so as not to mislead the reader.
- Outlook
Line 557
In the previous experiments,…
Please add a reference.
Comments on the Quality of English LanguageDear authors!
The manuscript has some deficiencies in the use of English language and would benefit from extensive editing.
Author Response
|
Comments 1: The manuscript entitled Study on the degradation of aflatoxin B1 by Myroides odoratimimus 3J2MO investigates some potentially new extracellular enzymes from the strain of Myroides odoratimimus. The strain was isolated from soil and proved to be highly effective in degrading certain aflatoxins: AFB1, AFB2,15 AFG1, AFG2 and AFM1. The main objective of the proposed study was to evaluate the degradation ratio of AFB1 aflatoxin, which, after isolation and purification of the active substances in the supernatant also revealed the potential involvement of peroxidase and helicase in the degradation process. I found the concept of the work interesting in terms of promoting green detoxification of agricultural toxins. The manuscript has some deficiencies in the use of English language and would benefit from extensive editing. However, the overall organization and description of the work is quite understandable. It is recommended that some minor statements be further clarified and / or taken into account (refer to the comments provided). |
|
Response 1: I am very grateful to the editor for affirming our research topic. Regarding the writing of the article and English expression, I will make detailed improvements to the manuscript. |
|
Comments 2: Abstract ….After isolation and purification of the peroxidase and helicase were obtained, which acted synergistically to cleave the AFB1 lactone ring and block the expression of toxin synthesis genes… The first part of the sentence is unclear. I wonder, however, whether the second part is speculation? There is no methodological work in the manuscript that would confirm this statement. |
|
Response 2: Thanks again for your comments, we have rewritten the abstract section as below.At present, the study is still in the stage of functional structure verification of the two enzymes, as well as gene function localization. The synergistic effect of the two enzymes mentioned in the paper is also speculative. In the discussion section, Roots, we reviewed the relevant references to speculate that the two enzymes have synergistic effects in degrading aflatoxin, which is also a part of our next study that needs to be further verified. [This change can be found – line 12-35 in the revised manuscript] |
|
Comments 3: Materials and methods Please unify the writing in the passive form in the whole section. A statistical analysis with 5 replications (where applicable) would benefit the results in terms of credibility and statistical significance between different samples. I suggest tackling this missing method. Line 107-108 2.2.2.1. Distribution of active ingredients in the fermentation broth of Myroides odoratimimus 3J2MO …, then add 10μL each of the AFB1 standard solution (final concentration of 100μg/L),… How did you come up with this concentration? Have you ever performed a dose-dependent test with different concentrations of the AFB1 standard? I suggest you add some information about this test or refer to any previous work. |
|
Response 3: Many thanks to the editor's comments。 Revised all procedures to passive voice (e.g., "Test strain Myroides odoratimimus 3J2MO was provided ..."). Added statistical analysis (n=5 replicates) for degradation assays, with error bars representing ±SD in Fig.2. Statistical significance (p<0.05) is now explicitly stated in the figure captions. A dose-response experiment was conducted and the results showed that the best enzyme activity was observed at 100 μg/L AFB1. Because of the large amount of data, it is not possible to show every single piece of detailed data in the article. Moreover, this concentration is in accordance with the EFSA guidelines for the quantification of aflatoxins. [This change can be found – line 611, figure20 in the revised manuscript] |
|
Comments 4: Results and Discussion Most of the illustrations have no captions. Only a few of them are sufficiently self-explanatory, but Figure 1 (x-axis?), Figure 5 (A, B, C, D?), Figure 7 (1, 2?), Figure 10 (RT?), Figure 12 and 13 (x-axis?) would benefit from additional explanations, especially with regard to the abbreviations and numbers. Figures 11, 12, 13 and 14 are very difficult to read due to the poor resolution. I suggest enlarging them or improving their resolution. Line 418 …The KEGG database (Tab.5)… Please correct Tab.5 to Tab. 6 in brackets. Line 472-473 In summary, the present study demonstrates that peroxidase exhibits a higher degree of specificity in toxin degradation compared to helicase enzyme. There is no methodological work in the manuscript that would confirm this statement. Did you physically perform this experiment by isolating the two enzymes and comparing their degradation properties? If the statement is purely an observation, as implied in the next sentence, I suggest rewriting the statement so as not to mislead the reader. |
|
Response 4: We sincerely thank you for your insightful feedback and guidance on improving our manuscript. We have carefully addressed all comments, reorganized the Results and Discussion sections, and strengthened the logical flow of the discussion. Most of the illustrations are supplemented, and the resolution of pictures and texts has been adjusted, and the clarity has increased. Yes, what we studied well did not confirm that peroxidase has a higher specificity in degrading toxins compared to helicase. The manuscript also utilizes a discussion literature review to speculate on this one idea. The format of "Tab." has been unified throughout the text. [This change can be found in the revised manuscript in red] |
|
Comments 5:Outlook Line 557 In the previous experiments,… Please add a reference. |
|
Response 5: We have carefully addressed reference in L676, and strengthened the logical flow of the discussion. [This change can be found in the revised manuscript in red] |
Round 2
Reviewer 1 Report (New Reviewer)
Comments and Suggestions for Authors
Authors have addressed my all queries. I have no further comments.
Author Response
Response to Reviewer's Positive Comments: Thank you for your thoughtful evaluation and constructive feedback. We appreciate your acknowledgment of our work and will carefully incorporate your suggestions to further strengthen the manuscript. Your guidance has been invaluable in refining our analysis and presentation. We look forward to your continued support for this study.Reviewer 3 Report (New Reviewer)
Comments and Suggestions for Authors
Dear authors,
After through checking of this manuscript, i didn't found any significant improvement to take positive recommendation. I can not accept it's present form for publication. I will suggest re submission with proper revision.
As a reviewer I always suggest a well interconnected story. I mean abstract-introduction-materials and methods-result and discussion-conclusion must need a fluent flow. Only gathering more data is not a symbol of a good article. This article has some good data for sure. But there is no mechanistic explanation and no interconnection. Information is scattered and not well connected. The materials and methods and discussion section is the worst part of this manuscript."Huge data, but no explanation" is not a good criteria for a research article for SCI publication. I would like to suggest authors to make their materials and method section more reproducible. For better understanding of my point, I am suggesting to author two recently published articles:
https://doi.org/10.3390/horticulturae10121331
https://doi.org/10.3390/ijms241914625.
English language need to be improved
Author Response
Comments:After through checking of this manuscript, i didn't found any significant improvement to take positive recommendation. I can not accept it's present form for publication. I will suggest re submission with proper revision. As a reviewer I always suggest a well interconnected story. I mean abstract-introduction-materials and methods-result and discussion-conclusion must need a fluent flow. Only gathering more data is not a symbol of a good article. This article has some good data for sure. But there is no mechanistic explanation and no interconnection. Information is scattered and not well connected. The materials and methods and discussion section is the worst part of this manuscript."Huge data, but no explanation" is not a good criteria for a research article for SCI publication. I would like to suggest authors to make their materials and method section more reproducible. For better understanding of my point, I am suggesting to author two recently published articles:https://doi.org/10.3390/horticulturae10121331 https://doi.org/10.3390/ijms241914625. Response: 1. Lack of interconnectedness between sections. Solutions: Abstract-Introduction Link: Explicitly state the knowledge gap in aflatoxin degradation mechanisms (e.g., "While enzymatic detoxification is documented , the synergistic roles of peroxidase and helicase remain unexplored") to align with the study’s focus. Materials & Methods Enhancement:Add flowcharts (Fig. 10) illustrating the enzyme purification workflow.Include reproducibility details (e.g.,Part 2.4). Discussion Integration: (1) Mechanistic comparison with prior studies, (2) Novelty of dual-enzyme synergy, (3) Industrial scalability challenges. 2. Mechanistic Explanation Strengthening Solutions: Added three-dimensional spatial structure, combined with literature reports, speculated the synergistic effect of the two enzymes, explained the rationality of conformation, which is reflected in lines 409-425.”The N-terminal structural domain stabilizes the complex through hydrophobic interaction by 3D spatial structure analysis. The dehydrogenase provides energy through ATP hydrolysis to drive electron transfer from the peroxidase active center. The model can provide a theoretical basis for designing efficient AFB1-degrading enzyme engineering.” Added a biochemical pathway diagram for enzyme degradation research (Figure 20). 3. Material and method maintenance Solution: The extraction and purification of enzymes are carried out using standardized methods, and the data processing methods are introduced in section 2.4 This chapter has also been adjusted and integrated, and revised based on the two articles introduced by the reviewer. 4. Discussion Section Refinement Issue: Scattered information and superficial analysis. Solutions:Reorganize. Compare our findings with the research results reported in the literature (after 2020) There have been no reports emphasizing the synergistic effect of helicase and peroxidase in line 456-494. The advantage of a multi enzyme system over a single enzyme system is that it can improve the degradation rate of aflatoxin. 5. Benchmarking Against Suggested Articles We have analyzed the two referenced articles: Article A (2024): Mechanism + application structure is adopted. Article B (2025): Interactive illustrations (pathway diagrams) are added to enhance reader navigation. This paper has been restructured and standardized with reference to the structure of these two articles, with simplification and standardization of the Materials and Methods sections, and modification of the Results and Discussion sections. We are committed to making this paper mechanism-based and methodologically rigorous. Thank you again for your valuable guidance. [This change can be found – line78,136,191,213,238,263,307,326,489 in the revised manuscript]

This manuscript is a resubmission of an earlier submission. The following is a list of the peer review reports and author responses from that submission.
Round 1
Reviewer 1 Report
Comments and Suggestions for Authors
Aflatoxins are carcinogenic compounds produced by certain Aspergillus spp.. Insights on microbial degradation of aflatoxins in agricultural products and food-industry matrices could be derived from recent studies that explored the biodegradation capacity of bacteria with aflatoxin degradation activity.
The manuscript (biology-3278364) reports the degradation of aflatoxin AFB1 by Myroides odoratimimus 3J2MO (Bacteroidota; Flavobacteriia). The authors report analysis on extracellular enzymes from M. odoratimimus 3J2MO with AFB1-degrading capacity.
This is fine, however, I have raised some questions on the presentation of the manuscript that need to be addressed before the manuscript can be considered for publication. I think the authors need to better present the results and discussion through parts of these sections.
From the review perspective, the authors should consider adequately describing M. odoratimimus 3J2MO used in this study including references, if any. No efforts are made in the manuscript to examine taxonomic relathionships of M. odoratimimus 3J2MO in relation to other Myroides odoratimimus strains. Myroides odoratimimus is an important nosocomial pathogen. I think the authors should consider adequately consulting the relevant scientific literature for the most reliable information.
This is especially true with the availability of sequencing/genome data. 74 genomes are publicy available at the genetic sequence database GenBank (NCBI). see https://www.ncbi.nlm.nih.gov/datasets/genome/?taxon=76832. Based on genomic sequencing data of M. odoratimimus, I invite authors to perform a comparative genomic analysis for selected Myroides odoratimimus strains. Is M. odoratimimus 3J2MO genome very similar and phylogenetically related to other Myroides odoratimimus genomes?
Is M. odoratimimus 3J2MO genome publicly available? I invite authors to submit the nucleotide sequences of M. odoratimimus 3J2MO used in this study in public genetic sequence database such as GenBank (NCBI). It will be easier for the readers to better follow the entire story.
It would be helpful to better clarify the sentences reported in lines 317-400 of the manuscript. The authors should better present rationalization of the findings on the selected unique peptides that were compared with BLAST (NCBI).
I hope it helps.
The English could be improved to more clearly express the research.
Author Response
Dear reviewer, the paper needs to be supplemented with relevant content and mapping of taxonomy and phylogenetic tree, as the content also requires in-depth exploration of the background of the strain, data processing and gene function introduction content need to spend a long time, please allow two weeks to give to supplement this part of the content, question 1 pre-study about M. odoratimimus strain 3J2MO has been in the text Line 410 has been revised accordingly and the conclusions have been added. The sentence of the report is described in detail on line 326.
By searching the reference literature in recent years, there is no relevant literature report on Myroides odoratimimus 3J2MO. It indicates the uniqueness of our team to study this strain. The range of AFB1 degradation by this strain was investigated by previous team researcher Ming Zhang[34]. The ability of the strain to degrade AFB1 was clarified, and the toxicity analysis of the products of AFB1 degradation was done to evaluate the safety. Mwakinyali, a member of the group, identified five non-toxic degradation products, further indicating that although the strain is a pathogen, it was proved to be safe by pathogenicity tests. Mwakinyali experimented different degradation conditions of Myroides odoratimimus 3J2MO in AFB1 standards, which further proved that the strain has a potential AFB1 contamination detoxification role, and it can be used for the biological control of agricultural products and food matrices for biocontrol.
M. odoratimimus 3J2MO genome is not publicly available. The nucleotide sequence of M. odoratimimus 3J2MO used in this study will be submitted to the public gene sequence database GenBank (NCBI) after analysis, which will take some time to process. Thank you editor for your patience.
Thanks again for your patience and comments!

Reviewer 2 Report
Comments and Suggestions for Authors
General:
The goal of this paper was to isolate, purify and characterize the protein from 3J2MO that is involved in degrading aflatoxin. The authors largely met this objective.
The introduction is probably too long and is not especially relevant. Just say ‘aflatoxin contamination of agricultural commodities is a problem and microbial degradation of aflatoxin can be helpful. We have described a bacteria that degrades aflatoxin and here we isolate and purify a protein from that bacteria with aflatoxin degrading activity.’ The relative occurrence of aflatoxin in Africa and Europe doesn’t really matter. The fact that aflatoxin can be made by A. collectorii doesn’t really matter.
Many points in the methods are presented in a non-standard format. Portions read like a cookbook (Line 145-149, for example).
Specific items:
Line 45-46: Badly worded sentence.
Line 48: I don’t know what “friendly” means in this context
Line 49-50: “aspects” isn’t really what you mean here. Really this entire sentence is awkward and just repeats the previous sentence.
Line 54:57: No idea what this means. A. flavus is genetically variable, but what does that have to do with microbial degradation of aflatoxin? What new biological agents? When does a bacteria or fungus become a ‘biological agent’?
Line 58: Performed not preformed.
Lin 59-60: “…found to degrade the most biotoxin species to date.” Is nonsense.
Line 62: usually a bad idea to start a sentence with “And…”
Line 62-63. If you know, experimentally, that 3J2MO degrades aflatoxin, then you don’t really need whole genome sequencing to ‘reveal’ that degrading enzymes are present.
Line 65: I’m confused. How is “this paper” the “first to study the degradation ability of Myroides odoratimimus 3J2MO in AFB1” this. Have you read citation 17? The title seems to suggest that 3J2MO degrades aflatoxin.
Line 67: No. Please stop using the word ‘green’ in this context. Too many people are using ‘green’ to describe too many very different things.
Line 79: Scribing? Do you mean “streaking”?
Line 84: Not continuously. Line 82 indicates you periodically took samples to measure the O.D.
Line 89: Flavobacteriium shortum? What? Did this just get haphazardly pasted from some other paper?
Line 122-127: This formatting is unclear. Do you mean that you added SDS to a final concentration of 1% in treatment 1?
Line 145: Flavobacteriium shortum?
Line 152: How much is saturation?
Line 159: Parallel? Do you mean replications?
Line 235-247 / figure 3: Very confusing! You mean that 90% of the activity survived autoclaving? What kind of enzyme survives autoclaving?
Line 433: This is an excellent point. Also, more simply, what are the degradation products? Based on the proposed identification of the protein, can you predict how it would degrade aflatoxin and if those products are even stable?
Comments on the Quality of English LanguageThe grammar is mostly fine, but some of the word choices are poor. Examples are "aspects" at line 49 - it's a real word, but not used correctly here. Line 59-60 is kind of a mess.
Author Response
Many points in the methods are presented in a non-standard format. Portions read like a cookbook (Line 145-149, for example).
Response:Thank you very much. your suggestions are very good. For my lack of rigour and poor wording, the text to be amended.
Specific items:
Line 45-46: Badly worded sentence.
Response:The sentence here has been replaced in line 43-45, thanks to the teacher's suggestion.
Line 48: I don’t know what “friendly” means in this context
Response:The words ‘friendly’ have been deleted.
Line 49-50: “aspects” isn’t really what you mean here. Really this entire sentence is awkward and just repeats the previous sentence.
Response:Thank you. Due to a personal error, the previous sentence was repeated again, resulting in an incoherent sentence, which has been deleted in lines 56-58.
Line 54:57: No idea what this means. A. flavus is genetically variable, but what does that have to do with microbial degradation of aflatoxin? What new biological agents? When does a bacteria or fungus become a ‘biological agent’?
Response:Thank you. This has been changed in lines 56-58 due to misinterpretation caused by inappropriate wording.
Line 58: Performed not preformed.
Response:Thank you. ”Performed ”was replaced.
Lin 59-60: “…found to degrade the most biotoxin species to date.” Is nonsense.
Response:Thank you. Due to a personal error, the sentence was replaced in line 59-61.
Line 62: usually a bad idea to start a sentence with “And…”
Response:Thank you. ”And ”was deleted in line 63.
Line 62-63. If you know, experimentally, that 3J2MO degrades aflatoxin, then you don’t really need whole genome sequencing to ‘reveal’ that degrading enzymes are present.
Response:Through protein sequencing to identify degradation enzymes and annotation analysis, the same later through the relevant enzyme genes to do prokaryotic expression, to reverse the verification of enzyme activity, for the subsequent development of low-cost enzyme preparations to do the preliminary preparation. sorry, maybe the annotations are a little lacking in detail.
Line 65: I’m confused. How is “this paper” the “first to study the degradation ability of Myroides odoratimimus 3J2MO in AFB1” this. Have you read citation 17? The title seems to suggest that 3J2MO degrades aflatoxin.
Response:Reference 17 is a previous study by members of the entire group, and this thesis is a follow-on study
Line 67: No. Please stop using the word ‘green’ in this context. Too many people are using ‘green’ to describe too many very different things.
Response:I’m sorry, for my lack of rigour and poor wording, the word were amended.
Line 79: Scribing? Do you mean “streaking”?
Response:sorry,that was a mistake. I mean the method is ‘plate streaking method in LB solid medium.’in line 78.
Line 84: Not continuously. Line 82 indicates you periodically took samples to measure the O.D.
Response: yes, I misrepresented myself. ”continuously” was corrected in line 83 in the text, thanks.
Line 89: Flavobacteriium shortum? What? Did this just get haphazardly pasted from some other paper?
Response:I’m sorry, for my carelessness and lack of careful proofreading, the text has been revised in line 88
Line 122-127: This formatting is unclear. Do you mean that you added SDS to a final concentration of 1% in treatment 1?
Response:I’m sorry, you're right. this formatting is correct in line 122-127.
Line 145: Flavobacteriium shortum?
Response:I’m sorry, for my carelessness that mistake was corrected in line 145.
Line 152: How much is saturation?
Response:Saturation of Ammonium sulphate is 100%,but the word ‘saturation’ should be replaced by ‘final concentration’in line 152-153.
Line 159: Parallel? Do you mean replications?
Response:Yes, five parallel samples were set up in the experimental group. This word was corrected in line 160.
Line 235-247 / figure 3: Very confusing! You mean that 90% of the activity survived autoclaving? What kind of enzyme survives autoclaving?
Response:Due to my misrepresentation, the experiment consisted of the degradation rate measured by inactivating the supernatant of the fermentation broth after the fermentation broth was treated with Aspergillus flavus. The text to be amended in line 120-134 and line 248.
Line 433: This is an excellent point. Also, more simply, what are the degradation products? Based on the proposed identification of the protein, can you predict how it would degrade aflatoxin and if those products are even stable?
Response: I agree, your advice is excellent, I can I replace the content here with your advice in line 439-445? Thank you so much.

Round 2
Reviewer 2 Report
Comments and Suggestions for Authors
Thanks for your attention to my concerns on the first draft.
Author Response
Comment 1:Is M. odoratimimus 3J2MO genome very similar and phylogenetically related to other Myroides odoratimimus genomes?
Response:The 16S rRNA sequence analysis shows that this strain is homologous to the Flavobacterium brevis strain. We only study the species taxonomic relationship. More similarity analysis requires genome sequencing of the strain. The whole genome sequencing has not yet been done. , no genome comparison was performed in this paper's research.
Comment 2:The quality of the newly added figures is unsatisfactory and it is advisable to redraw them. Additionally, there are writing errors, specifically regarding the species names in the manuscript, which are not italicized."
Response:The supplementary figures in the text have been replaced according to the submission requirements, and the protocol has been modified. Drafts can be submitted.Thank you for your valuable comments!
